# Closing the policy gap in children and youth suicide in low- and middle-income countries: The case for mental health policies

## Perspective

suicide; child; youth; mental health; children and youth mental health; policy

**Corresponding author:**
Abir Aldhalimi;
Email: Abir.aldhalimi@yale.edu

## Abir Aldhalimi [ORCID]

School of Medicine, Yale University, USA

## Abstract

Suicide is the third leading cause of death among children and youth aged 10–24, and nearly three-quarters of suicides occur in low- and middle-income countries (LMICs). Despite global efforts, children and youth mental health and suicide prevention remain underprioritized in national policy and are often deployed separately in LMICs. Governments should develop standalone, multisectoral mental health policies for children and youth that integrate suicide prevention strategies and that address social determinants of suicide risk. This commentary aims to inform national policymakers, global health and international development actors, and researchers engaged in the mental health and suicide prevention of children and youth in LMICs.

## Impact Statement

Suicide is the third leading cause of death among people aged 10–24, and nearly three-quarters of suicides occur in low- and middle-income countries (LMICs). Structural and social factors, including poverty, violence, marginalization, unemployment, unregulated social media use and digital harm, increase distress and vulnerability to suicide. Yet, despite recognition by the Lancet Commission on Adolescent Health and Wellbeing (2025) and the World Health Organization (2025a), national children and youth mental health and suicide prevention policies remain scarce, fragmented, underfunded, often deployed separately, and do not address social factors (Patel et al., 2018). This commentary argues that governments should develop standalone, multisectorial children and youth mental health policies that embed suicide prevention efforts, are rights-based, and address the social determinants that increase suicide risk (Pirkis et al., 2024). These efforts require collaboration across national governments, global health and international development actors, and researchers.

## Introduction

Children and youth aged 10–24 account for a substantial share of the mental health burden in low- and middle-income countries (LMICs; Gore et al., 2011; World Health Organization, 2022). Suicide is the third leading cause of death among this age group, and nearly three-quarters of suicides occur in LMICs (World Health Organization, 2025a). This commentary, consistent with global health policy frameworks, including those developed by the World Health Organization (2021) and the Lancet Commission on Adolescent Health and Wellbeing (2025), focuses on children and youth within this age range.

Children and youth differ developmentally, socially, and psychologically from adults and face distinct mental health and suicide risks and barriers to care. For example, children and youth are more likely to experience violence, bullying, and harm associated with social media use. Further, children and youth may encounter obstacles to health care access related to consent, privacy, and confidentiality, especially when seeking services without a parent or caregiver. Their unique experiences underscore the need for standalone mental health policies.

The crisis of children and youth suicide in LMICs is a policy emergency as much as a public and mental health emergency, and governments play an important role in addressing these challenges. Children and youth mental health policies and suicide prevention efforts are rare in LMICs. When governments do employ them, they are often executed separately. This siloed approach undermines the sustainability of these well-intentioned efforts. Suicide is complex, and prevention efforts are more likely to be effective and sustainable when they are embedded within mental health policies that are multisectoral. Consistent with the World Health Organization's Guidance on Mental Health Policy and Strategic Action Plan (2025b), these policies should also

address social determinants, use a rights-based perspective, and deploy a whole-of-government approach supported by sustainable funding.

Prior studies have found that suicide in LMICs is associated with structural, social, and economic stressors (Iemmi et al., 2016; Knipe et al., 2019), especially when mental health care is limited. These stressors include exposure to violence and war, civil unrest, potentially traumatic experiences, poverty, marginalization, and discrimination based on identity, loneliness, and the use of social media (Barry et al., 2013; United Nations Children's Fund, 2021; Kirkbride et al., 2024). This points to the fact that mental health policies and prevention strategies must go beyond the health sector and be situated in economic, education, democracy, and human rights sectors.

## Policy gap

Children and youth mental health is underprioritized in national policies, despite calls by major actors in the global mental health space, including the Lancet Commission call for large-scale investment (Lancet Commission on Adolescent Health and Wellbeing, 2025; Patel et al., 2018). While the World Health Organization's *Mental Health Atlas* (2024) reported that nearly half of the surveyed countries had a children and youth mental health policy or plan, it does not distinguish between standalone policies and plans that are integrated into adult mental health initiatives, making it difficult to assess progress in standalone children and youth policies. Even where policies exist, implementation is limited by scarce human and financial resources.

Suicide prevention efforts are even less developed. Only 6% of low-income countries (as defined by the World Bank's income classification system) that responded to the World Health Organization's data call reported having an existing suicide prevention strategy, and none had programs in active implementation (World Health Organization, 2024). In lower-middle-income countries, 37% of the respondents noted having a national plan, and 36% reported an actively implemented program. While an improvement, this average is still well below that of upper-middle-income countries (51% have a strategy, and 49% have a plan) and high-income countries (67% have a strategy, and 73% have a plan; World Health Organization, 2024). In other words, the countries with the highest rates of children and youth suicide have yet to prioritize suicide prevention. Moreover, few LMICs tailor their suicide prevention efforts for young people.

These findings highlight that mental health policies or plans are often established separately from suicide prevention strategies in LMICs. Suicide is often recognized as a public health crisis, and efforts to address suicide are often led by either non-governmental organizations or governmental health departments (Vijayakumar and Phillips, 2016; Pirkis et al., 2024). While there are benefits to departments of health leading such efforts, including having dedicated funding, designating one sector as the sole leader of a suicide prevention program risks a lack of integration into broader, more sustainable mental health systems.

Further, children and youth most often interact with professionals outside the health sector, such as those in schools and communities. Therefore, a comprehensive approach to mental health care should include efforts to strengthen peer support and peer educators, train teachers and community mental health workers, and increase access to evidence-based digital mental health services (Galagali and Brooks, 2020).

## Policy examples

Sri Lanka was able to decrease the suicide rate by 70% between the 1990s and 2010s by restricting access to pesticides (Knipe et al., 2017). Although this approach has been successful, it does not address the factors that increase suicide risk, such as economic and social hardships, or create mental health systems for early detection or continuity of care. Recent reporting by the World Health Organization (2025c) documents youth in Sri Lanka continuing to struggle with suicide in high numbers, with 15% of youth having considered attempting suicide in the last year. Further, 1 in 10 reported attempting suicide, according to self-reported survey results. The World Health Organization's reporting highlights the complex social, health, and economic challenges youth face in the country. Further, Sri Lankan mental health services are underfunded nationally, especially for young people, which may impact mental health workforce development (Vijayakumar and Phillips, 2016). While the government has been attempting to introduce a more modern, rights-based mental health law since 2005, mental health policy reform has not been successful. The country's current mental health law focuses largely on detention or involuntary hospitalization of those with mental illness rather than community-based care (Hapangama et al., 2023).

Sri Lanka's approach presents possibilities and limitations of national suicide prevention strategies centered on means restriction. While pesticide control is one of the most effective population-level suicide prevention strategies, emerging evidence suggests that comprehensive suicide strategies require a multisectoral approach. According to the World Health Organization's Live Life framework (2021), national suicide prevention strategies for children and youth need a layered approach that includes responsible media reporting, promotion of socio-emotional life skills, and early identification, assessment, management, and follow-up. These interventions are strengthened by the policy framework outlined in the World Health Organization's mental health policy guidance (2025b). The success of national suicide prevention strategies will depend on effective governance, robust financing, workforce development, multisectoral interventions, and rights-based service delivery.

In contrast, Australia's standalone children and youth mental health policy may serve as a helpful model. Australia's policy includes strong legal protections, age-appropriate and rights-based care, early detection, workforce development, and clear accountability across government branches. It works alongside their National Mental Health and Suicide Prevention Plan (2025–2035). In this plan, Australia not only establishes a holistic, system-wide approach to suicide prevention, but it also addresses social and structural factors that increase suicide risk, including financial hardship, housing, and social inclusion (Australian Government National Mental Health Commission, 2025). The approach is based on human rights principles and employs education-sector policies, child-protection frameworks, school-based and community programs, and clinical services (Hung et al., 2025).

## Policy considerations

Governments should formally recognize the intertwined nature of children and youth mental health, suicide prevention, and social

and economic factors. This recognition can begin with the establishment of standalone, multisectoral, rights-based mental health policies designed specifically for children and youth. Standalone policies would strengthen governance while operating within national mental health systems and working across sectors. Taking recommendations from the Lancet Commission on Adolescent Health and Wellbeing (2025), the World Health Organization (2025b), and the United Nations Children's Fund (2021), policies should prioritize early detection, prevention, continuity of care, and rights-based service delivery.

Comprehensive suicide prevention can be embedded within these policies, with interventions that include public education, restricting access to means of suicide (e.g., pesticides), and the decriminalization of suicide attempts (World Health Organization, 2021). Interventions should also include crisis intervention services, community-based care models, and clear referral pathways to specialized mental health providers (World Health Organization, 2013; Vijayakumar and Phillips, 2016). In line with the World Health Organization's Guidance on Mental Health Policy and Strategic Action (2025b), dedicated budgets and long-term financing plans must back these policies (Zhou et al., 2020; World Health Organization, 2021) and be designed and implemented across various government sectors. Progress should be tracked through quantifiable performance indicators with clear accountability plans (United Nations Children's Fund, 2021).

These broad policies should also address social determinants, especially those related to economic hardship and poverty, such as minimum wage and unemployment, to reduce the financial burdens that contribute to mental health distress and suicide risk. Policies that reduce discrimination and protect human rights, strengthen social cohesion and reduce loneliness, foster social support, and implement school-based and workplace wellness initiatives are also vital to reducing the risk of mental health distress and suicide (Kirkbride et al., 2024; Pirkis et al., 2024).

To ensure policies reflect the realities and needs of communities, the policy development process needs to include meaningful engagement with children, youth, and families. Children and youth should be engaged at all stages of the policy development process to ensure solutions reflect their lived experience and foster trust in mental health systems (Lancet Commission on Adolescent Health and Wellbeing, 2025; Patel et al., 2018). The design of these policies must involve diverse stakeholders from the community, non-government, and private sectors, engaged in youth mental health and suicide prevention efforts, ensuring a comprehensive societal approach (Vijayakumar and Phillips, 2016).

It is well established in the literature that there is a workforce shortage of specialists in children and youth mental health and in suicide response. Mental health policies should therefore prioritize training and expanding this workforce. Efforts might include creating pathways for professional specialization and training non-specialist providers, such as teachers, community health workers, and primary care staff, to deliver basic psychosocial support and to identify early signs of distress (World Health Organization, 2013; Zhou et al., 2020).

Lastly, digital technology can be an accessible and low-cost intervention to provide mental health services for this age group. Research shows that phone or mobile programs can increase access to mental health care by limiting barriers, such as transportation and stigma. Specific to suicide prevention, digital tools can assist with early detection through screening, provide psychoeducation, offer crisis support by connecting children and youth with helplines or chat services, and provide referral to care (Galagali and Brooks, 2020).

## The role of global actors

National governments and global development actors should cooperate to close the gap in this critical policy area. International development actors, including the World Health Organization, United Nations Children's Fund, and the World Bank, as well as implementation science researchers, are positioned to provide effective support to governments. There is a critical shortage of technical assistance and research on mental health policy frameworks and their implementation, especially on mental health policies that include comprehensive suicide prevention initiatives. Global actors are positioned well to strengthen support for national policy development by offering technical guidance to governments, creating strategies to shore up workforce development, elevating youth voices in governance, and building the capacity of mental health ministries and departments, and local non-governmental organizations (NGOs) that advocate for youth mental health and suicide prevention (Patel et al., 2018; United Nations Children's Fund, 2021).

Zhou et al. (2020) conducted a systematic review of youth mental health policies in LMICs and found that limited public awareness of the importance of youth mental health was associated with limited political will among governments to prioritize this issue. Competing priorities, including other public health challenges, place youth mental health at a severe disadvantage. For example, governments have spent approximately 2% of their health budgets on mental health globally (World Health Organization, 2024).

Global actors can move beyond advocacy and position children and youth mental health and suicide prevention as a foundational pillar for human capital development and economic growth. Untreated mental health conditions impede productivity and long-term development (World Health Organization, 2016). In fact, scaling up mental health care, especially for depression and anxiety, results in a fourfold return on investment due to improved productivity and health outcomes (Chisholm et al., 2016). Global actors can increase local government support by leveraging economic data to mobilize funding and to prioritize these issues.

Finally, effective policy development requires collaboration among researchers, international development actors, local NGOs, and governments. Aligning technical assistance with evidence-based policy frameworks, financing mechanisms, and community engagement is vital for effectively translating policy to real progress in children and youth mental health and suicide prevention.

## Conclusion

Mental health and suicide prevention in children and youth are public health and policy emergencies. Evidence shows that suicide in LMICs is often associated with social and economic strain, exposure to violence and marginalization, and limited access to mental health care. As such, suicide prevention strategies are most effective when embedded in broad mental health policies that extend beyond clinical care, adopt cross-sectoral, human rights, and community-based approaches, and address social determinants that contribute to suicide risk. Governments can be supported by international development actors, donors, and researchers through technical assistance, evidence-based children- and youth-specific frameworks, community consultations and partnerships, and sustained financing.

In addition to policy commitment, this area is ripe for research. Future research should examine how standalone children and youth mental health policies can be effectively designed and

implemented in LMICs. Strengthening these efforts can help inform policy, reduce suicide risk, and improve the mental health of children and youth in LMICs.

**Open peer review.** To view the open peer review materials for this article, please visit http://doi.org/10.1017/gmh.2026.10168.

**Data availability statement.** No new data were generated or analyzed in support of this research.

**Author contribution.** Abir Aldhalimi, PhD, developed and wrote the manuscript.

**Financial support.** This work received no funding.

**Competing interests.** The author declares no conflicts of interest.

**Ethics statement.** Ethical approval was not required for this article, as it presents a perspective based on previously published literature.

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
