## [Reviewer Report]

This is a timely and well argued paper. There are one or two grammatical glitches.

I would suggest somewhat modifying the call for stand alone youth mental health policies that include suicide prevention so that it is clear that such stand alone policies will also need to be integrated into each of the relevant sector policies eg general health, social welfare, education, criminal justice, employment etc. Otherwise stand alone mental health policies tend not to be well implemented!

I suggest including a box that gives all the usual components of a national suicide prevention strategy, referencing UN/WHO guidance.

I somewhat disagree with the distinction the authors make about suicide in LMIC being more associated with social risk factors and less with mental illness than in HIC. In the studies I am familiar with, both psychiatric disorders and social risk factors are equally relevant in all countries.

---

## [Reviewer Report]

This perspective focuses on how child and youth suicide is inadequately addressed by governmental bodies and national policies in LMIC and suggests, in broad strokes, some reforms that could lead to improved youth suicide prevention in these countries. The author urges LMIC governments, international development partners, and researchers to create dedicated, “whole of-government” mental health policies that address child and youth suicide prevention directly, expand the care workforce, involve young people, and secure sustainable funding and technical support. Overall, the perspective is well-considered and identifies key policy gaps.

I hope the below points will be useful to the author if they would like to refine the perspective.

I suggest clarifying who is the primary audience for this perspective (paper).

The perspective should explicitly state the age group it addresses. Although the manuscript sometimes refers to “children” and at other times to “youth,” it is unclear whether the intended scope includes all individuals under 18, under 24, or the narrower 10‑24 year range mentioned in instances. Clarifying this point would help readers understand whether the perspective applies to the entire child/youth population or solely to the “youth” age range typically recognized by many governments and intergovernmental bodies. That typical youth age range excludes younger children (5‑9 years), who face a non‑zero risk of suicide and could benefit from targeted policy reforms.

Regarding the case‑study section, policymakers unfamiliar with suicide prevention need a clear takeaway. While the examples illustrate successful pesticide‑control campaigns in LMIC, emerging evidence also supports a variety of other preventive approaches—clinical interventions, school‑based education, and community programs. The concluding paragraph of Section 2.2 would be more informative if it summarized the available intervention modalities or levels that might constitute a comprehensive set for a national strategy.

The case studies are limited to two Asian countries, yet LMIC outside of Asia have national strategies. Focusing exclusively on geographically adjacent Asian examples narrows the perspective’s global relevance and may give the impression that those examples and the overall perspective are not more broadly applicable.

Finally, given the rapid expansion of digital technologies and mental‑health applications worldwide, the policy considerations would benefit from a discussion of the potential of digitally based preventive interventions. What is the authors view on how national strategies might leverage technology and digital media for child/youth suicide prevention? If such integration is deemed inappropriate, what is the rationale?

The conclusion claims there is a “clear roadmap” for guiding LMIC toward “integrated, sustainable, youth focused systems.” While the perspective highlights LMIC successes—such as pesticide control initiatives—and outlines useful policy considerations, labeling these examples as a roadmap (i.e., a detailed plan) overstates the extent of the guidance that can be offered, particularly in regions where evidence, money, and political will are most lacking.

---

## [Reviewer Report]

Thank you for the opportunity to review this important manuscript. This manuscript highlights an issue of central importance for mental health and for adolescent development and wellbeing. The author’s argument is that stand-alone suicide prevention policies are needed specifically for children and adolescents. Additional points raised are the importance of youth involvement and the finding that many youth suicides occur outside the context of experiencing a mental health condition.

The piece could be strengthened in a number of ways. First, the central argument is the need for standalone policies for children and adolescents that focus on suicide prevention and related issues. As a reader, I am left with the question of why standalone policies have greater likelihood of being successful? It would help to describe in more detail what countries have standalone policies. The author cites that more than half of countries worldwide have child and adolescent mental health policies. What are the benefits that have been observed in these countries vs. the other half of the world’s countries that lack such policies? Have the countries with such policies shown greater suicide reduction compared to those without policies? Or are there other indicators, such as those governments with standalone policies have a) greater funding for mental health; b) more child and adolescent mental health providers; or c) some other indicator? It would help to do an analysis of items in the WHO Atlas comparing those countries who do and do not have standalone policies. Overall, on a superficial level, the argument for standalone policies makes sense, but the commentary should go deeper to actually demonstrate that there are some types of positive results of such policies.

Another issue is that the case studies are very brief and not particularly helpful in making the author’s point. From the case studies, it sounds like India has a policy but Sri Lanka does not? Yes, in Sri Lanka, greater reductions were seen because of the pesticide control measures. The author makes the case that pesticide measures are insufficient to generate benefit. However, if one were to compare countries with and without pesticide measures, I am guessing that there would be substantial differences in reduction of suicide rates. Therefore, making the case that standalone policies are the most important step may not hold up if all countries adopted pesticide control measures comparable to Sri Lanka and other countries.

The author raises a very interesting point that suicides often occur outside the context of mental health conditions. There are a number of studies that support this. However, this seems at odds with the demand for stand alone child and adolescent mental health policies. If many of the deaths occur outside the context of mental health conditions, then would an emphasis on mental health policies be insufficient to tackle key causes and factors. The author addresses this by calling for more approaches to social determinants. This seems logical, but outside the purview of an exclusive mental health policy.

Finally, the language used in the commentary is inconsistent with the evidence. The use of ‘must’ in many sentences comes across as demanding something that may still require further exploration. The ‘must’ language is also at odds with some of the statements about youth-led and youth-collaborative approaches—youth may decide that other approaches are needed than standalone policies. The repeated ‘must’ language also comes across as potentially patronizing and colonial from an American institution author directed at the global community.

In summary, this is an extremely important topic. The advocacy for standalone policies is definitely worth exploring. However, considerable more detail about why standalone policies are currently working and would be more successful than other approaches is lacking. Additionally, softening the tone to be something that is more collaborative and in dialogues with governments and youth communities worldwide would be more constructive.

Thank you again for the opportunity to read this important piece.

---

## [Editor Report]

Dear Abir, thanks for submitting this perspective for the special issue on self-harm and suicide.

We asked three external reviewers to comment on your manuscript and all agree that the issue is highly relevant but that more nuance and clarity would strengthen this piece.

I hope you’re willing to address their comments as this would be a great addition to our special issue!

Best, Jer

---

## [Reviewer Report]

I thank the author for their time in preparing the revision and addressing my comments and suggestions. Good work.

---

## [Reviewer Report]

I am content with the author’s response to the reviewer comments.

I have spotted these glitches in the text .

Abstract

Line 25 Insert missing word here “these policies need to XXXXX structural and social factors””

Intro Line 52 ? insert “when they are embedded WITHIN mental health policies that are multisectoral”

---

## [Editor Report]

I have reviewed the most recent edits made to the manuscript and verified that the author has addressed the pending issues.